# Comparison of efficacy and safety of single versus repeated intra-articular injection of allogeneic neonatal mesenchymal stem cells for treatment of osteoarthritis of the metacarpophalangeal/metatarsophalangeal joint in horses: A clinical pilot study

**Carmelo Magri[1]ᵒ\*, Michael Schramme[1]ᵒ, Marine Febre[2]ᵒ, Eddy Cauvin[3]‡, Fabrice Labadie[2]‡, Nathalie Saulnier[2]‡, Isé François[1]‡, Antoine Lechartier[4]‡, David Aebischer[5]‡, Anne-Sophie Moncelet[6]‡, Stéphane Maddens[2]ᵒ**

**1** Clinéquine, VetAgro Sup, Campus Vétérinaire de Lyon, Marcy l'Etoile, France, **2** Vetbiobank SAS, Campus Vétérinaire de Lyon, Marcy l'Etoile, France, **3** Azurvet, Cagnes-sur-Mer, France, **4** Clinique équine vétérinaire de Meheudin, Ecouché, France, **5** Clinique équine Areda, Bex, Switzerland, **6** Clinique équine de Grosbois, Marolles-en-Brie, France

ᵒ These authors contributed equally to this work.
‡ These authors also contributed equally to this work.
\* vet.magri@gmail.com

## Abstract

The purpose of this prospective study was to evaluate the effects of single and repeated intra-articular administration of allogeneic, umbilical cord-derived, neonatal mesenchymal stem cells (MSC) in horses with lameness due to osteoarthritis (OA) of a metacarpophalangeal joint (MPJ).

Twenty-eight horses were included. Horses were divided into two groups. Horses in group MSC1 received an MSC injection at M0 and a placebo injection at M1 (1 month after M0). Horses in group MSC2 received MSC injections at M0 and at M1. Joint injections were performed with a blinded syringe. Clinical assessment was performed by the treating veterinarian at M1, M2 and M6 (2 and 6 months after M0), including lameness evaluation, palpation and flexion of the joint. Radiographic examination of the treated joints was performed at inclusion and repeated at M6. Radiographs were anonymized and assessed by 2 ECVDI LA associate members. Short term safety assessment was performed by owner survey. A 2-month rehabilitation program was recommended to veterinarians. There was a significant improvement of the total clinical score for horses in both groups. There was no significant difference in the total clinical score between groups MSC1 and MSC2 at any time point in the study. There was no significant difference in the total radiographic OA score, osteophyte score, joint space width score and subchondral bone score between inclusion and M6. Owner-detected adverse effects to MSC injection were recorded in 18% of the horses. Lameness caused by OA improved significantly over the 6-month duration of the study after treatment with allogeneic neonatal umbilical cord-derived MSCs combined with 8 weeks

**Data Availability Statement:** All relevant data are within the paper and its Supporting Information files.

**Funding:** Azurvet provided support in the form of salary for author EC. Vetbiobank provided support in the form of salaries for authors (MF, FL, NS) and contributed to the study design, data collection and analysis, decision to publish and preparation of the manuscript. The specific roles of these authors are articulated in the 'Authors Contribution Section'.

**Competing interests:** The authors have read the journal's policy and the authors of this manuscript have the following competing interests: SM is a current and principal shareholder of Vetbiobank. MF, FL, NS, are employees of Vetbiobank. There is no commercial or financial connection between Vetbiobank and the seven co-authors (CM, MS, EC, IF, AL, DA, AM). EC is shareholder of Azurvet. The product used in this study is developed for commercial use in France and Europe. There are no patents associated with this research to declare. These commercial affiliations do not alter adherence of all authors to all PLOS ONE policies on sharing data and materials. MS and CM are current employees of the National Veterinary School Of Lyon, Vetagro Sup in France.

rest and rehabilitation. There is no apparent clinical benefit of repeated intra-articular administration of MSCs at a 1-month interval in horses with MPJ OA when compared to the effect of a single injection.

## Introduction

Osteoarthritis (OA) of the metacarpophalangeal/metatarsophalangeal joints (MPJs) is one of the most common causes of lameness in Sports horses [1]. Several local and systemic treatments have been described, including intra-articular viscosupplementation (hyaluronic acid, polyacrylamide gel), anti-inflammatory biological therapeutics [platelet-rich plasma (PRP), autologous conditioned serum (ACS or IRAP), polysulphated glycosaminoglycans (PSGAGS) and steroidal drugs (corticosteroids, stanozolol). Such intra-articular treatments have been shown to have predominantly symptom-modifying effects.

Mesenchymal stem cell (MSC) therapy has been developed as a means of promoting scar-free tissue regeneration in a variety of musculoskeletal injuries in horses [2], and has been characterized predominantly for the treatment of overstrain injuries of the superficial digital flexor tendon [3–5]. Even if the direct regenerative properties of MSCs are now being questioned [6], their immunomodulatory effects may offer an alternative mechanism to improve the quality of the healing process of injured tissues through disease-modifying effects.

MSCs have been used in equine joints for the treatment of osteoarthritis [7] and for surgical resurfacing of chondral defects during arthroscopic interventions [8–10]. Although the results of intra-articular injection of MSCs were initially disappointing in the management of OA [11], more promising results have since been reported [8,12,13]. Preclinical data in a rabbit model have shown that equine MSCs may be able to prevent cartilage degradation after induced meniscal injury [14].

MSCs can be harvested from different sources, and autologous preparations derived from bone marrow or adipose tissue were the first to be introduced in equine orthopaedics [4,11,15]. The preparation and culture of autologous cells is time-consuming and their properties can be variable depending on the individual horse donor. Therefore the therapeutic use of allogeneic MSCs derived from bone marrow, adipose tissue, peripheral blood, gingiva, and other tissues from selected adult donors has also been explored [16–20]. More recently, umbilical cord, umbilical cord blood or placenta have gained acceptance as sources of neonatal allogeneic MSCs because these cells may have greater biological potential and an immunologically privileged state compared to adult MSCs [21]. Several studies argue for a relatively higher immunomodulatory potential of these cells both in horses and other species, justifying their investigation for application in joints to help improve the clinical status of the joint through balancing the inflammatory environment [14].

As the persistence of injected MSCs in the joint is limited[14,22,23], their therapeutic effect is likely to be temporary. A single intra-articular injection of MSCs can therefore no longer be considered as an 'OA treatment for life' and repeated intra-articular dosing may have more beneficial effects. The optimal time for a first MSC-injection has been shown to be during a period of joint inflammation [14] but optimal times and dosages for repeat injections still need to be defined. For a pharmacological drug, optimal time for dosing is usually derived from drug metabolism and pharmacokinetic (DMPK) studies. However, MSCs have unconventional DMPK parameters which furthermore are not completely understood [24]. As a consequence, MSC dosing regimens have been derived from experience and approximation.

Repeated administration of adult MSCs has raised concerns related to allogenicity [25], but neonatal umbilical cord MSCs may be better tolerated than adult allogenic MSCs [26].

The purpose of this study was to evaluate the effects of single and repeated intra-articular administration of allogeneic, umbilical cord-derived, neonatal MSCs in horses with lameness due to documented OA of an MPJ. We hypothesized that 2 intra-articular MSC injections with a 4-week inter-injection interval would improve the clinical and radiological outcome parameters at 6 months compared to a single intra-articular injection.

The primary objectives of this study were to determine the 1) clinical and 2) radiological outcomes of horses with OA of an MPJ treated with intra-articular MSCs and 3) to compare the outcomes between horses that were injected once or twice. The secondary objective was to assess the safety of single and repeated intra-articular administration of allogeneic neonatal MSCs in equine MP joints.

## Materials and methods

### Study design (Fig 1)

This was a multicenter, randomized, double-blinded, controlled, clinical pilot study, involving veterinary surgeons from 10 different equine veterinary hospitals in France. All horses were client-owned and all owners signed an informed consent prior to inclusion. The study was implemented in accordance with University regulations. All horses were sedated during joint injections and all efforts were made to minimize suffering. Horses were divided into two groups (MSC1 and MSC2) according to the number of MSC injections received. The study compared a single MSC injection (group MSC1) with 2 MSC injections (group MSC2). Horses in group MSC1 received an MSC injection at M0 and a placebo injection at M1 (1 month after M0). Horses in group MSC2 received an MSC injection at M0 and at M1. Follow-up examinations were performed at M1, M2 (2 months after M0) and M6 (6 months after M0). Owners and attending veterinarians were blinded to the selected treatment regimen for each horse during the 6-month duration of the study. Short term safety assessment was performed by owner survey using dedicated forms. At M6, after the final lameness evaluation, treatment group allocation was unveiled to both owners and veterinarians.

### Inclusion and exclusion criteria

Horses were selected for the study if they had a persistent single-limb lameness of at least 2 months' duration that was localized to a MPJ with diagnostic analgesia, that was exacerbated by a fetlock flexion test and if radiographs showed osteoarthritis of that MPJ.

Clinical examination included lameness evaluation, palpation and flexion of the joint. Lameness was scored from 0 to 5 based on an adaptation of the AAEP (American Association of Equine Practitioners) lameness grading system for trotting on a straight line only (0 sound; 1 lameness difficult to detect and inconsistent; 2 lameness difficult to detect but consistent; 3 lameness consistently observable in a straight line; 4 obvious lameness with marked head nodding; 5 minimal weight-bearing in motion and/or at rest) [27]. When available, objective gait

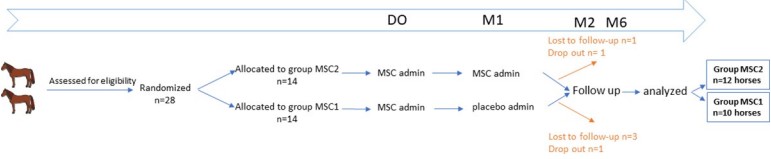

**Fig 1. Case selection and time-line of the study.**

analysis was performed with an inertial sensor-based lameness diagnosis system (Lameness Locator ND) while horses were trotted on concrete in straight lines. The parameters used included mean HDMax (Maximum head height difference), mean HDMin (Minimum head height difference) and Vector Sum for fore limb lameness, and mean PDmax (Maximum pelvic height difference) and mean PDmin (Minimum pelvic height difference) for hindlimb lameness.[28] Additional scores were given for the horse's response to fetlock flexion at rest (passive flexion 0–4 = no, slight, moderate, marked, severe pain response), for the increase in lameness grade following a fetlock flexion test (active flexion 0–4 = no, slight, moderate, marked, severe lameness increase) and for joint distention (0–4 = no, slight, moderate, marked, severe distention). The four clinical scores (lameness, passive flexion response, active flexion response and joint distention) were added to obtain a composite clinical score out of a total of 17.

Radiographic examination consisted of 4 projections (dorso-palmar/plantar, latero-medial, dorsolateral-palmaromedial oblique and dorsomedial-palmarolateral oblique views). Magnetic resonance imaging (MRI), ultrasonography and diagnostic arthroscopy were also used in some horses for identification of cartilage degeneration and osteophytes.

Previously attempted treatments were recorded as well as the time of initial diagnosis of OA. Horses with other orthopaedic injuries and horses that received intra-articular treatment less than 1 month prior to the MSC injection were not included. Any other intra-articular treatment given during the 6-month duration of the study was a cause for exclusion from the study, as was the incidence of any other orthopaedic injury that could have influenced lameness evaluation of the treated limb during the follow-up period. Previous treatment with MSCs was also a cause for exclusion. Systemic administration of anti-inflammatory medication was accepted during the study if it occurred within 48 hours of or more than 4 weeks after MSC injection and was reported in the follow-up information. Administration of any anti-inflammatory medication was not allowed less than 15 days before any of the clinical evaluations (M0, M1, M2 and M6).

### Cell product preparation

Equine umbilical cord-derived (UCd)-MSCs were provided by Vetbiobank (Marcy l'Etoile, France). Umbilical cords were recovered following parturition from selected mares and foals belonging to French National Studfarms (IFCE, Saumur, France), in accordance with a previously described protocol [29]. Extensive adventitious agent screening (including viruses, parasites and mycoplasma) was performed on mares' blood samples before foaling and on neonatal tissue samples afterwards. All batches of MSCs used in this study were isolated from umbilical cord tissue, cultured, manufactured and characterised as previously described [14]. Briefly, cells displayed a conventional phenotype for equine MSCs (i.e. CD44+; CD29+; CD90+; MHC1+; MHC2-; CD45-; CD34-), differentiated *in vitro* into adipogenic and osteogenic cell lineages and expressed chondrogenic markers upon differentiation [30]. MSCs were frozen in 10% (v/v) DMSO and stored in liquid nitrogen.

On the day prior to injection, each MSC dose was retrieved from liquid nitrogen, thawed at 37°C and washed extensively with Dulbecco's Phosphate Buffered Saline[a] (D-PBS, Pan Biotech). The investigational product consisted of approximately $10 \times 10^6$ viable cells (median: 11 $\times 10^6$; minimum: $9 \times 10^6$ and maximum $16 \times 10^6$) at the $4^{th}$ passage (minimum 2, maximum 8 passages), re-suspended after thawing in 2 mL D-PBS and transferred to a sterile 5mL ready-to-use syringe. The placebo product consisted of syringes containing the same volume of D-PBS to ensure a similar color and consistency of content compared to MSC-loaded syringes. Blinding was further assured by camouflaging the syringe's content with opaque adhesive

sterile tape applied around the barrel of the syringe. Loaded syringes were shipped overnight, refrigerated between 2 to 12°C in an appropriately sterile container, to equine hospitals participating in the study and used within 12 hours of reception. These conditions assure less than 10% cell death at the time of injection.

Eleven different horse donors were used to prepare equal numbers of MSC batches. It was mandatory for horses in group MSC2 that the cell batch used for the second injection was from a different donor horse than the one used for first injection. Major Histocompatibility Complex (MHC) typing was not performed in this study.

### Treatment (Fig 1)

Joint injections were performed at M0 (MSCs) and M1 (MSCs or placebo) under sedation with an alpha-2-agonist (detomidine[b] 0,01 mg/kg IV, xylazine[c] 0,06 m/kg IV or romifidine[d] 0,08 mg/kg IV) alone or in combination with butorphanol[e] (0,01 mg/kg IV). The injection procedure was identical for both MSC and placebo syringes. Prior to injection both MSC and placebo syringes were gently inverted 8–10 times to bring MSCs into homogeneous suspension. The injections were performed slowly, over a minimum duration of 7–10 seconds, using 20-gauge needles to prevent cell damage from shear stress [31].

The use of systemic anti-inflammatory medication with phenylbutazone[f] (2,2 mg/kg IV), flunixine meglumine[g] (1,1 mg/kg IV) and/or dexamethasone[h] (0,1 mg/kg IV) immediately before injection was considered acceptable to avoid potential post-injection reactions. Immediately after injection a standard lower limb bandage was placed on the limb and maintained for 24 hours.

### Rehabilitation

A 2-month rehabilitation program was recommended to veterinarians (Fig 2) consisting of stall rest with short, progressively increasing, periods of controlled hand-walking daily (from 15–30 minutes). After 2 months horses were free to resume their normal exercise routine. Deviations from the proposed rehabilitation protocol had to be reported by the horse's treating veterinarian to the study monitor.

### Follow-up evaluation

Clinical assessment was performed by the treating veterinarian at M1, M2 and M6 as for the inclusion. Radiographic examination of the treated MPJ was repeated at M6. Radiographs of both studies were anonymized and assessed by 2 ECVDI LA associate members (EC, MS). An OA score was given to each radiographic exam (4 views) using a customized scoring system (based on subchondral bone changes, joint space width, and periarticular new bone formation) to obtain a total radiographic score out of 10 (Table 1). Owners were asked to document in writing any post-injection reactions at the level of the injected MPJ during the first 48 hours after each injection and to send the completed form to the stem cell laboratory (Vetbiobank).

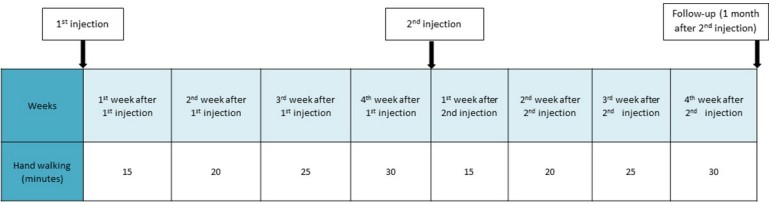

**Fig 2. Rehabilitation program.** The 2-month rehabilitation program consisted of stall rest with periods of controlled hand walking daily. After 2 months horses were free to resume their normal exercise routine.

**Table 1. Radiographic score for fetlock osteoarthritis.**

| A. Subchondral bone / epiphyseal bone changes | 0: no significant changes,<br>1: diffuse sclerosis, subchondral bone thickening,<br>2: focal or marginal subchondral bone lysis, cyst-like subchondral lesions<br>3: diffuse or multifocal subchondral bone lysis |
|---|---|
| B. Articular space changes | 0: no significant changes,<br>1: mild space collapse,<br>2: moderate space collapse,<br>3: severe space collapse,<br>4: ankylosis |
| C. Periarticular new bone formation (osteophytes and enthesophytes) | 0: no significant changes,<br>1: Mild remodeling of articular margins<br>2: Moderate remodeling with spur-like formations<br>3: Severe to exuberant new bone formation |
| Total OA score = | A + B + C |

Radiographic grading system used to calculate a cumulative fetlock total osteoarthritis (OA) score

## Statistical analysis

Statistical analysis was performed with the GraphPad Prism 6.0 program[i]. Baseline comparisons between groups (MSC1 and MSC2) for age, clinical scores, and radiographic scores were performed using the Mann-Whitney U test. Categorical data (gender) were compared using Fisher's exact test and limb affected (front versus hind limb) was compared using a $\chi 2$ test. The clinical and radiographic scores of horses from each group and from both groups combined at different time points (M1, M2 and M6), were compared to baseline (M0) using the non-parametric Friedman repeated measures ANOVA on a rank sum test. Post hoc comparisons were then performed using a Dunn's test.

Clinical scores were compared between groups (MSC1 and MSC2) using a 2-way ANOVA corrected by Sidak's test. Radiographic scores were compared between groups using Wilcoxon's matched pairs test.

Differences were considered significant for P values $\leq$0.05.

## Results

Twenty-eight horses were included in the study.

Six horses were excluded from the study during the 6-month follow-up period. One horse died of colic 3 weeks after the first MSC injection (H24). Horse H28 developed lameness in the contralateral MP joint with radiographic signs of OA. The horse received additional treatments that led to exclusion from the study. This case led us to amend protocol adding a systematic radiographic examination of the contralateral MPJ prior to horses' acceptance to the study. Horse H23 was excluded after the second MSC injection resulted in an articular 'flare' reaction within 24 hours that required a joint lavage and un-blinding of the treatment for this horse. Three horses were lost to follow up (H25, H26, H27) because they were not presented for follow-up evaluations.

A total of twenty-two horses (78,5%) completed the 6-month follow-up period, 10 in group MSC1 and 12 in group MSC2.

Median age at inclusion was 9 years [range 2–17] with 2 stallions, 12 geldings and 8 mares included. Breeds included 15 Warmbloods (9 Selle Français, 2 BWP, 1 SBS, 1 KWPN, 1 Swiss Warmblood, 1 Oldenburg, 2 Standardbreds, 1 Paint horse, 1 Lusitano, 1 Anglo-arabian, 1 mixed breed and 1 Arabian). Sixteen horses were used for jumping, 1 for dressage, 1 for

**Table 2.  Clinical and radiographic scores for group MSC1, group MSC2 and both groups combined.**

| | MSC1 (median & range) | MSC2 (median & range) | P-value Difference MSC1-MSC2 | All horses (median & range) | P- value compared to M0 |
|---|---|---|---|---|---|
| | **Total Clinical scores** | | | | |
| M0 | 8 [4 -14] | 8 [4 -12] | 0.34 | 8 [4 -16] | |
| M1 | 5 [0 -14]* | 5 [1 - 9] | 0.54 | 5,5 [0 -14] * | 0,0021 |
| M2 | 6 [2 - 13]* | 5 [0 - 7]* | 0.15 | 5 [0 -13] * | 0,0003 |
| M6 | 3.5 [0 - 15]* | 3.5 [0 - 10]* | 0.33 | 3,5 [0 -15] * | < 0,0001 |
| | **Lameness scores (median)** | | | | |
| M0 | 2 | 1.5 | 0.48 | 2 | |
| M1 | 1* | 1 | 0.98 | 1* | 0,0105 |
| M2 | 1* | 1.5 | >0.99 | 1* | 0,0017 |
| M6 | 1* | 0.5 | 0.62 | 1* | 0,0017 |
| | **Joint distension scores (median)** | | | | |
| M0 | 2 | 1.5 | 0.94 | 2 | |
| M1 | 1.5 | 1 | 0.98 | 1 | 0,185 |
| M2 | 1 | 1 | 0.99 | 1 | 0,079 |
| M6 | 1 | 0 | 0.86 | 0,5* | 0,021 |
| | **Passive Flexion scores (median)** | | | | |
| M0 | 2.5 | 2.5 | 0.98 | 2,5 | |
| M1 | 2 | 1 | 0.42 | 1* | 0,007 |
| M2 | 1.5 | 0.5* | 0.45 | 1* | 0,003 |
| M6 | 1* | 0* | 0.75 | 0,5* | 0,001 |
| | **Active Flexion scores (median)** | | | | |
| M0 | 2 | 3 | 0.99 | 3 | |
| M1 | 1.75 | 2 | 0.99 | 2 | 0,05 |
| M2 | 2 | 1* | 0.86 | 2* | 0,004 |
| M6 | 1 | 1* | 0.99 | 1* | 0,001 |
| | **Radiographic OA scores (median & range)** | | | | |
| M0 | 3.5 [2–6] | 2 [1 -7] | 0.46 | 2,25 | |
| M6 | 5 [2.5 - 6.5] | 2.5 [1 -7.5] | 0.41 | 3,25* | 0,0020 |
| | **Radiographic osteophytes scores (median)** | | | | |
| M0 | 2 | 1 | 0.28 | 1,75 | |
| M6 | 2.5 | 1.5 | 0.17 | 2 | 0,175 |
| | **Radiographic joint space width scores (median)** | | | | |
| M0 | 1 | 0 | 0.52 | 0,25 | |
| M6 | 1.5 | 0.5 | 0.42 | 0,5* | 0,016 |
| | **Radiographic subchondral bone scores (median)** | | | | |
| M0 | 0.5 | 0 | 0.88 | 0,5 | |
| M6 | 0.5 | 0.5 | 0.94 | 0,5 | 0,316 |

Table 2 shows all clinical scores and sub-scores (median and range) for horses in group MSC1, group MSC2 and both groups combined (all horses) at M0, M1, M2 and M6 and all radiographic scores and sub-scores (median and range) at M0 and M6. Significance was set at p<0.05.

* indicates a significant difference with the measurement obtained at M0.

pleasure riding, 1 for western riding, 1 for endurance, and 2 for Standardbred racing. There were 16 fore limb and 6 hind limb MPJs.

Clinical and radiographic scores are summarized in Table 2. At inclusion, there were no significant differences between groups MSC1 and MSC2 for age (p = 0,7587), fore limb vs hind limb location (p = 0,7587), lameness score (p = 0,21), passive flexion score (p = 0,78), active

flexion score (p = 0,95), joint distension (p = 0,59), total clinical score (p = 0,6360) and radiographic score (p = 0,46). There were more females in group MSC1 than group MSC2 (p = 0,0062). Nineteen of the 22 horses had previously received other treatments which had failed to improve lameness. These treatments had included stall rest (n = 10), intra-articular injection of corticoids (n = 5) hyaluronic acid (n = 4), IRAP or PRP (n = 3), intravenous administration of tiludronate (n = 3), oral NSAID (n = 1), chondroprotective nutritional complements (n = 1) and arthroscopic debridement and joint lavage (n = 9). Of the 9 horses that were included after a diagnostic arthroscopy, 4 ended up in the MSC1 group and 5 in the MSC2 group.

The mean delay between diagnosis of OA and MSC treatment was 481 days (median 356 days; range 89–2573) in the 11/22 horses for which this information was available. Six of these horses had not been used in equestrian sports or racing for at least 12 months preceding MSC treatment.

At the time of first MSC injection, prophylactic intravenous anti-inflammatory medication consisting of an NSAID alone (flunixin or phenylbutazone) was administered to 10/28 horses, an NSAID (flunixin or phenylbutazone) with dexamethasone to another 10/28 horses and dexamethasone alone to 1 horse. Five horses did not receive any anti-inflammatory medication and the information was missing for 2 horses. At the time of the second injection at M1, 11/28 horses received an NSAID/dexamethasone combination, 7 horses only an NSAID, and 1 horse only dexamethasone. Four horses did not receive any anti-inflammatory medication and the information was missing for 4 horses. During the 6 months of the study, only 3 horses received additional oral NSAIDs: 2 because of persistent joint effusion and pain (one at 2 days and one at 15 days after injection) and another one for pain in the contralateral limb at M4.

A total of 4/34 (12%) MSC injections and 2/10 (20%) placebo injections resulted in owner-reported adverse effects in 4 of 28 horses (H1, H2, H17, H23). H1 developed mild periarticular heat for 48 hours after the first injection and heat and joint distension after the second injection (placebo) for 48 hours. H2 developed joint distension after the first injection but not after the second injection (MSCs). H17 had no reported adverse effect after the first MSC injection but developed mild periarticular heat for 48 hours after the second injection (placebo). Only one horse (H23) experienced adverse effects at both injection times. This horse received two MSC injections and developed marked articular and periarticular swelling with a grade 4/5 lameness after the second injection (M1). No prophylactic anti-inflammatory medication had been administered to this horse. Cytology revealed a leukocyte concentration of 103,000 cells/μL with 70% neutrophils but bacteriological culture of the synovial fluid was negative. This prompted the attending practitioner to perform a through-and-through joint lavage under standing sedation 24 hours after MSC injection. Strikingly all 4 horses with reported adverse effects had a positive outcome. H23 was free of lameness, joint effusion or flexion responses at M6 (total clinical score improved from 6 to 0) but was excluded from the study. No other adverse effects were reported by the owners during a one-year follow up period. There was no difference in the incidence of adverse effects between horses that received and horses that did not receive prophylactic anti-inflammatory injections (p = 0.92). None of the 23 joint injections that were preceded by a prophylactic intravenous dexamethasone injection resulted in any reported adverse effects. No correlation could be established between any characteristics of each cell batch (cell viability, culture passage numbers and total viable cell number) and the occurrence of reported adverse effects.

Clinical scores were available for 22 horses (Table 2). There was improvement of the clinical score in 9 out of 10 horses from group MSC1 and in all 12 horses from group MSC2. There was a significant improvement of the total clinical score for horses in group MSC1 between M0 and M1 (p = 0,0361), M0 and M2 (p = 0,0128) and M0 and M6 (p = 0,0030). There was

also significant improvement of the total clinical score for horses in group MSC2 between M0 and M2 (p = 0,0342) and between M0 and M6 (p = 0,0008). However, there was no significant difference in the total clinical score between groups MSC1 and MSC2 at any time point in the study: M0 (p = 0,6792), M1(p = 0,6254), M2 (p = 0,3378) or M6 (p = 0,3087). When considered individually for both groups and for all horses, lameness scores, flexion tests and joint effusion were all significantly reduced at M2 and M6 (and lameness score and passive flexion also at M1), and there were no differences between both groups (Table 2). Objective gait analysis was available at M0 and M6 for 6 horses, 5 with fore limb lameness (H4, H7, H8, H20, H22) and one with hind limb lameness (H21). There was improvement in the objective lameness parameters (vector sum, max diff head/pelvis and min diff head/pelvis) in 3 out of 6 horses, which was in agreement with the subjective assessment for 5 of these horses. Radiographic scores were available for 16 horses (Table 2), 5 horses from group MSC1 and 11 horses from group MSC2. There was no significant difference in the total radiographic OA score, osteophyte score, joint space width score and subchondral bone score between M0 and M6 (p = 0,5597) in either group MSC1 or group MSC2, and there were no differences between both groups.

At the end of the study, 5 horses returned to their intended level of use, 8 to a lower level and 9 remained lame. Of the 19 horses who were involved in competitive equestrian sports before inclusion in the study, 13 (68%) resumed competition (12 showjumpers, 1 endurance) while 6 (32%) did not (5 racehorses that failed to race; 1 horse developed contralateral limb tendinitis). Three horses were included in the study at the insistence of their owners without any work-related objectives but in order to improve the horses' well-being by reducing lameness.

## Discussion

Several studies have investigated safety and efficacy of single or repeated intra-articular injections of allogeneic adult MSCs for the treatment of OA in horses ([8,11,13,17,32]). Whereas studies have documented the safety of equine neonatal MSCs [26], thereby confirming results obtained in other species [33,34], their clinical efficacy has not been investigated neither following a single nor repeated injections in horses. This study therefore, is the first to compare clinical outcomes of two different treatment regimens using intra-articular injection of allogeneic neonatal MSCs in horses with MPJ OA.

The study was designed as a multicentered (n = 10) trial, involving both academic and private practices, to minimize the risks of repeated bias related to inclusion, treatment or evaluation that may occur in cell therapy studies limited to one or two participating clinical centers. Biological products, including cell therapies, are inherently variable from one batch to another. To reduce the impact of such variability, clinical studies with allogeneic cells are generally performed with a single batch of cells to run the entire experiment [32,35] Such practice introduces a batch-specific result which would translate poorly into clinical practice where cells from different batches derived from different donors are necessarily used. Therefore we deliberately used several different batches of cells in our study, in order to reproduce better the reality of clinical practice and increase the clinical relevance of the study. Similarly, anticipating true clinical practice conditions where donor/patient MHC typing and matching would be difficult to implement, cells were not MHC-typed, and cells from a different donor than used for the first injection were used for the second MSC injection. Few equine MSC studies have been performed in a blinded fashion, which introduces a high risk of bias when subjective clinical outcome parameters are used, like lameness grading and scoring of flexion test responses. Different methodologies to implement blindness can be used. In a study performed by Broeckx *et al.*, the practitioner evaluating clinical outcome was different to the one administering the

treatment [32]. In agreement with investigation sites in our study, this method was considered difficult to implement and masking of the injectate was chosen instead.

The assessment of horses' return to athletic performance was based on public performance databases (letrot.com; francegalop.com; ffe.com) rather than relying on owners' assessment.

Owner-detected adverse effects to MSC injection were recorded in 18% of the horses in our study, three horses after the first injection and one horse after the second injection. Studies performed with adult autologous MSCs have shown self-limiting local inflammation after a single intra-articular injection of MSCs combined with fetal bovine serum (FBS) or hyaluronic acid [8,36]. In one study, the concurrent use of hyaluronic acid was incriminated for a 9% incidence of joint flares after injection of femorotibial joints with autologous bone marrow-derived MSCs[8].

A recent study explored the safety of a single intra-articular injection of adult allogeneic peripheral blood-derived MSCs in healthy horses [37]. There was transient joint heat and lameness after injection but no differences were found between the treated group and a control group of healthy horses undergoing the same injection protocol. In another study performed by the same group, evaluating efficacy of the same allogeneic peripheral blood-derived MSCs in horses with fetlock osteoarthritis, no adverse effects were observed after a single intra-articular injection. The authors included prophylactic administration of a systemic dose of an NSAID, which may have led to an underestimation bias of adverse effects [32].

When using allogeneic MSCs in repeat injections, specific immune responses against MHC antigens may occur. Induced cytotoxic antibodies to donor MHC antigens have been described against adult allogeneic MSCs in horses [38,39]. Both inflammatory stimuli present in the OA environment and cellular differentiation, have been shown *in vitro* to upregulate MHC-II antigen expression by MSCs [40]. Therefore, the lack of detectable expression of MHC-II antigens in cell batches, without priming, is not a definitive assurance against mounting effective immune reactions. However neonatal MSCs have been shown to express less MHC-II than their adult counterparts under inflammatory conditions [21]. Regardless, other authors have not been able to show any difference in post-injection reaction between autologous and allogeneic MSCs, whether the cells where neonatal [26] or bone marrow-derived adult cells [41]. Yet another study, evaluating neonatal MSCs in dogs with OA, failed to identify a consistent specific humoral response after single or repeated MSC injection [34]. The use of cells from different donors and the removal of fetal calf serum residues by extensive washing of cell cultures may further help reduce the risk of a local inflammatory reaction after a second MSC administration. The adverse effects observed in 2 horses after placebo injection (20%) suggests the possibility of non-specific inflammation occurring related to the arthrocentesis procedure itself or to the potential hyperreactive state of OA joints.

Only one horse in our study experienced a severe inflammatory reaction following injection of MSCs, resulting in severe swelling, heat and lameness (H23). Synovial fluid analysis (103,000 white blood cells/μL) was indicative of joint sepsis, possibly related to a breach in aseptic technique. Even so, this increased cell number might also be compatible with the presence of the injected MSCs in the joint fluid one day after injection or with an immune-mediated non-septic inflammatory synovitis ('joint flare'), especially as clinical signs of joint sepsis usually take longer than 24 hours to appear. Unfortunately, no cytological evaluation of the joint fluid was performed to determine whether the elevated cell count was caused by the presence of neutrophils, monocytes, lymphocytes or MSCs. Consequently, and even though a bacteriological culture remained negative, a conclusive diagnosis as to the cause of the acute inflammatory joint reaction was not possible.

It is important to note that the occurrence of adverse injection effects was not related to a negative outcome in our study. All 4 horses had a favorable outcome and H23 had returned to

his previous performance level at 6 months. This observation, though anecdotal, is similar to a previous report that suggested that horses with marked injection reactions had generally an excellent outcome compared to others [42]. Interestingly this also agrees with findings from a study revealing a link between a pro-inflammatory response early after MSC administration and immunomodulatory functions of MSCs [43]. An inflammatory reaction observed after intra-articular injection of MSCs, that is controllable medically with NSAIDs, may need to be considered more as an expected side effect rather that an adverse reaction.

The results of our study failed to show an improved outcome after two MSC injections with a 1-month injection interval, in comparison with a single MSC injection. The mechanism of action of MSCs is currently thought to be immunomodulatory one rather than any regenerative effects brought about by the MSCs' differentiation into chondrocytes [21]. The immunomodulatory effects of MSCs are thought to alter the inflammatory environment in the joint after joint injury through an anti-inflammatory activity of the MSCs as well of cells recruited to the injury site by the MSCs through their paracrine effect [16]. It would therefore be reasonable to expect a more profound or persistent beneficial anti-inflammatory effect from two administrations of MSCs than from a single administration. However, we were unable to show this. Several reasons may explain the apparent lack of benefit of a second MSC injection. Group sizes may have been too small with regard to the large variation in lameness scores to detect a significant difference between both groups. Also, the interval of 1 month between both injections may not be adequate. It was empirically chosen to fit with the estimated residence time of MSCs in the joint [22,44]. The first injection could have resulted in a humoral response with optimal antibody levels peaking at 1 month, leading to a rapid elimination of the MSCs of the second injection. It is further possible that the effect of the second injection was 'diluted' by the course of action of the first injection in other ways. As the mechanism of action of MSCs is still incompletely understood, their effects may not merely be cumulative as could be expected in a classical pharmacological dose-response effect of repeated intra-articular drug administration. For example, if inflammatory parameters of the joint were lowered by the first injection, MSCs from the second injection may not have encountered the expected threshold of inflammatory stimuli to be primed for optimal anti-inflammatory activity [14,32]. Therefore, it would be useful to test a longer inter-injection interval such as 6 months, as suggested by recent studies [45]. Beneficial effect of a second intra-articular injection of equine umbilical cord-derived MSCs with a 6-month inter-injection interval have been demonstrated in a pilot study with canine OA patients[34].

It is striking to note that horses in both groups experienced a significant improvement in lameness scores and total clinical scores, as early as M1, and over the entire duration of the study (M6). This finding is in agreement with those from efficacy studies of intra-articular administration of adult MSCs for the treatment of OA in horses [8,12,13,32,17]. A recent study, Broeckx et al. managed to include a placebo group in a study evaluating clinical efficacy of an intra-articular combination therapy with MSCs and equine allogeneic plasma. Unfortunately the authors used a saline control injection for the placebo group, which prevents a definitive distinction between the relative efficacy of the MSCs and the allogeneic plasma each in their own right [32]. When comparing our outcome data with the saline placebo group from this study, horses from both groups (MSC1 and MSC2) in our study had a significantly better outcome (p = 0,019). Even though our results compare favorably with outcome data from other MSC efficacy studies, a definitive conclusion about the purported beneficial effect of equine neonatal allogeneic umbilical cord-derived MSCs would require a prospective study including a group of horses with MPJ OA treated with an intra-articular placebo injection and subjected to the same period of rest and rehabilitation. However, given the nature of clinical practice, it is almost impossible to convince horse owners to collaborate with a 6-month study

in which there is a real chance that their horse will miss out on a potentially useful new therapeutic modality or drug.

The 8-week period of rest and rehabilitation may also have helped the horse's lameness grade improve through the study. It has been shown that it may be difficult to differentiate the beneficial effects of a rest and rehab program from those caused by a therapeutic agent administered concurrently in the treatment of equine osteoarthritis [46]. However, the authors feel this is unlikely, as at least 7 horses had already been rested for periods exceeding 8 weeks without improvement before recruitment in the study, other horses had failed to respond to other treatments combined with rest, and the median time between OA diagnosis and MSC treatment was one year with a minimum of 3 months for horses in this study.

There were no significant changes in the radiographic scores over the duration of the study. Other studies using models of equine OA also failed to show a significant change in radiographic signs of OA between MSC- and placebo-treated joints [40,11]. Furthermore, it is important to note there was a large variation of the total radiographic OA score of horses included in the study, ranging from mild to severe OA. This could have limited the ability to recognize a statistically significant improvement in OA scores at 6 months after MSC injection (M6). Even so, radiographic changes of OA like osteophytes, joint space narrowing and subchondral bone lysis tend to be irreversible and would not be expected to improve with any treatment. Therefore, the absence of significant exacerbation of the OA grade in either group over a period of 6 months could be considered as a positive finding.

Our study had some limitations inherent to equine multicenter clinical field trials. An increasing number of investigational centers induces variability in clinical assessment and uneven data collection. Objective lameness evaluation was only feasible in a limited number of patients due to limited availability of the Lameness locator equipment[j]. Not all horses were evaluated after intra-articular analgesia of the MPJ to confirm the presence of intra-articular pain associated with osteoarthritis. Even so, we feel that the combination of perineural analgesia, a positive flexion test and imaging findings can be conclusive for a diagnosis of osteoarthritis, even without the benefit of a recorded response to intra-articular analgesia [1]. Drugs selected for prophylactic treatment for prevention of adverse injection effects varied between operators. Inconsistency in radiographic protocol meant that all views required for grading the OA score were only present in 16 horses (MSC1 = 5 and MSC2 = 11). And finally, during the 6 months of the study, some horses were lost to follow up due to non-compliance of the owners. The low number of 22 horses retained for the final analysis could consequently induce the risk of a type II statistical error.

We conclude that lameness caused by MPJ OA improved significantly after treatment with allogeneic neonatal umbilical cord-derived MSCs combined with 8 weeks' rest and rehabilitation over the 6-month duration of the study. Intra-articular administration of umbilical cord-derived, allogeneic, neonatal MSCs carries a low risk of unexpected adverse injection effects. There is no apparent clinical benefit of repeated intra-articular administration of MSCs at a 1-month interval in horses with MPJ OA when compared to the effect of a single injection.

## Manufacturers' addresses

a Pan Biotech Gmbh, Aidenbach, Germany
b Detogesic, Zoetis, Malakoff, France
c Rompun Bayer Healthcare, Loos, France
d Sedivet, Merial, Lyon, France
e Torbugesic, Zoetis, Malakoff, France
f Phenylarthrite, Vetoquinol, Lure, France

g Finadyne Intervet, Beaucouze, France
h Dexadreson Intervet, Beaucouze, France
i Graphpad software, San Diego, USA
j Lameness Locator, Equinosis, Columbia, USA

## Supporting information

**S1 Table. Description of the population.**
(DOCX)

**S2 Table. Clinical scores of the horses.**
(XLSX)

**S3 Table. Radiographic scores of the horses.**
(XLSX)

## Acknowledgments

We are grateful to C. Dubois and L. Wimel, the staff of the experimental farm of the French Horse and Riding Institute (IFCE), 19370 Chamberet, for providing neonatal tissues from selected mares and foals. We thank Dr. K. Pader, Dr JM. Betsch, Dr M. Hamon, Dr E. Oua-chée, Dr A. Vitte, Dr N. Delalande, Dr F. Croisier, Dr JC. Meunier, Dr D. Lavorel, Dr T. Bertholdy, Dr F. Martin and Dr H. Sorribas for providing patients and performing clinical evaluations and data collections. We wish to thank Dr S. Sage for his contributions to the study protocol.

## Author Contributions

**Conceptualization:** Marine Febre, Fabrice Labadie, Nathalie Saulnier, Stéphane Maddens.

**Data curation:** Carmelo Magri, Marine Febre.

**Formal analysis:** Marine Febre.

**Funding acquisition:** Stéphane Maddens.

**Investigation:** Carmelo Magri, Eddy Cauvin, Isé François, Antoine Lechartier, David Aebischer, Anne-Sophie Moncelet.

**Methodology:** Marine Febre, Fabrice Labadie, Nathalie Saulnier, Isé François, Stéphane Maddens.

**Project administration:** Stéphane Maddens.

**Resources:** Carmelo Magri, Marine Febre, Isé François.

**Supervision:** Stéphane Maddens.

**Validation:** Stéphane Maddens.

**Visualization:** Marine Febre.

**Writing – original draft:** Carmelo Magri, Michael Schramme, Marine Febre.

**Writing – review & editing:** Carmelo Magri, Michael Schramme, Marine Febre, Stéphane Maddens.

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
