## [Decision Letter · Decision Letter 0]

6 Aug 2019

Comparison of efficacy and safety of single versus repeated intra-articular injection of allogeneic neonatal mesenchymal stem cells for treatment of osteoarthritis of the metacarpophalangeal/metatarsophalangeal joint in horses: a clinical pilot study

PONE-D-19-16694

Dear Dr. Magri,

We are pleased to inform you that your manuscript has been judged scientifically suitable for publication and will be formally accepted for publication once it complies with all outstanding technical requirements.

With kind regards,

Paolo Fiorina, MD, PhD

Academic Editor

PLOS ONE

JOURNAL REQUIREMENTS:

Please address the following in your final submission:

1. Thank you for including your competing interests statement; "I have read the journal's policy and the authors of this manuscript have the following competing interests:

SM is a current and principal shareholder of Vetbiobank. MF, FL, NS, are employees of Vetbiobank. There is no commercial or financial connection between Vetbiobank  and the seven co-authors (CM, MS, EC, IF,AL,DA, AM)"

We note that one or more of the authors are employed by a commercial company: Azurvet, Clinéquine and Vetbiobank SAS

Please respond by return email with an updated Funding Statement and Competing Interests Statement and we will change the online submission form on your behalf.

Reviewers' comments:

Reviewer's Responses to Questions

**Comments to the Author**

1. Is the manuscript technically sound, and do the data support the conclusions?

Reviewer #1: Yes

Reviewer #2: Yes

2. Has the statistical analysis been performed appropriately and rigorously? 

Reviewer #1: Yes

Reviewer #2: Yes

3. Have the authors made all data underlying the findings in their manuscript fully available?

Reviewer #1: Yes

Reviewer #2: Yes

4. Is the manuscript presented in an intelligible fashion and written in standard English?

Reviewer #1: Yes

Reviewer #2: Yes

5. Review Comments to the Author

Reviewer #1: The clinical study was correctly set up, the procedures were well explained and the results correctly presented

Reviewer #2: In this work authors describe the benefit of MSC based therapy as an approach to cure OA and MPJ in horses. The horses were divided in two group depending on the number of injections received. Authors of this clinical pilot study evaluated the effects of a single/repeated intrarticular administration of allogenic, umbilical cord-derived, neonatal MSCs on osteoarthritis (OA) of a

metacarpophalangeal joint (MPJ) both in term of efficacy (clinical or radiological) than of safety. Results showed that there was no difference between the two group of treatment, but as authors said in the discussion sessions the article is limited by the small numerosity of the cohort.

The study is very well designed and written and it has very interesting results providing important informations both for basic science and clinical science.

6. PLOS authors have the option to publish the peer review history of their article (what does this mean?). If published, this will include your full peer review and any attached files.

Reviewer #1: No

Reviewer #2: No

---

## [Editor Report · Acceptance letter]

22 Aug 2019

PONE-D-19-16694 

Comparison of efficacy and safety of single versus repeated intra-articular injection of allogeneic neonatal mesenchymal stem cells for treatment of osteoarthritis of the metacarpophalangeal/metatarsophalangeal joint in horses: a clinical pilot study 

Dear Dr. Magri:

I am pleased to inform you that your manuscript has been deemed suitable for publication in PLOS ONE. Congratulations! Your manuscript is now with our production department. 

With kind regards,

on behalf of

Dr. Paolo Fiorina 

Academic Editor

PLOS ONE